# Neuroprotection in Glaucoma: Basic Aspects and Clinical Relevance

**DOI:** 10.3390/jpm12111884

**Published:** 2022-11-10

**Authors:** Che-Yuan Kuo, Catherine Jui-Ling Liu

**Affiliations:** 1Department of Ophthalmology, Taipei Veterans General Hospital, Taipei 11217, Taiwan; 2Faculty of Medicine, School of Medicine, National Yang Ming Chiao Tung University, Taipei 11221, Taiwan

**Keywords:** glaucomatous optic neuropathy, glutamate, optic nerve protection

## Abstract

Glaucoma is a neurodegenerative disease that affects primarily the retinal ganglion cells (RGCs). Increased intraocular pressure (IOP) is one of the major risk factors for glaucoma. The mainstay of current glaucoma therapy is limited to lowering IOP; however, controlling IOP in certain patients can be futile in slowing disease progression. The understanding of potential biomolecular processes that occur in glaucomatous degeneration allows for the development of glaucoma treatments that modulate the death of RGCs. Neuroprotection is the modification of RGCs and the microenvironment of neurons to promote neuron survival and function. Numerous studies have revealed effective neuroprotection modalities in animal models of glaucoma; nevertheless, clinical translation remains a major challenge. In this review, we select the most clinically relevant treatment strategies, summarize preclinical and clinical data as well as recent therapeutic advances in IOP-independent neuroprotection research, and discuss the feasibility and hurdles of each therapeutic approach based on possible pathogenic mechanisms. We also summarize the potential therapeutic mechanisms of various agents in neuroprotection related to glutamate excitotoxicity.

## 1. Introduction

Glaucoma is one of the leading causes of irreversible blindness worldwide; it is an optic neuropathy characterized by the progressive loss of the visual field due to the apoptosis of retinal ganglion cells (RGCs) [1]. It is a multifactorial disease with complex pathogenesis that is not yet fully understood (Figure 1).

Intraocular pressure (IOP) is one of the most important risk factors for the development and progression of glaucoma, and IOP-lowering therapy is widely regarded as the only effective treatment strategy for slowing down or halting the deterioration of glaucomatous optic neuropathy. Reductions in IOP can be achieved with medication, laser, or surgery. Most patients with glaucoma, even after laser or surgical treatment, are treated with topical ocular hypotensive medications, which work by reducing the production of aqueous humor or facilitating the trabecular or uveoscleral aqueous outflow. These anti-glaucoma medications include β-adrenergic antagonists; α2-adrenergic agonists; carbonic anhydrase inhibitors; prostaglandin F2a analogs; and, more recently, Rho kinase inhibitors, latanoprostene bunod, and omidenepag isopropyl. In individuals with normal tension or high-tension glaucoma, ocular hypotensive medications are useful in delaying or preventing disease progression [2,3,4].

Although there are a variety of hypotensive medications and surgical techniques that can efficiently and effectively lower IOP, IOP reduction is sometimes insufficient to prevent glaucoma progression. In the Ocular Hypertension Treatment Study, 4.4% of medicated participants developed glaucoma 5 years after follow-up despite a 22.5% IOP reduction from an average of 24.9 mmHg to 19.3 mmHg [4]. Disease progression also occurred in 45% of treated patients who had a 25% IOP reduction from baseline IOP of 20.6 mmHg in the Early Manifest Glaucoma Trial [3]. Moreover, an even lower IOP does not preclude the possibility of glaucoma progression, as the Collaborative Normal-Tension Glaucoma Study revealed that 12% of treated patients experienced disease progression despite a 30% reduction to an average IOP of 10.6 mmHg during 5.6 years’ follow-up [5]. In addition, previous evidence has indicated that glaucoma is primarily an optic neuropathy with the optic nerve head (ONH) being the primary site of the disease [6,7]. As a result, more and more ophthalmic researchers have paid attention to investigating biomolecular mechanisms behind neuronal survival and developing further neuroprotective therapies as a supplement to IOP-lowering treatment.

Neuroprotection is a therapeutic approach that aims at preserving neural structure and function [8]. In glaucoma, neuroprotection refers to non-IOP-related interventions that can prevent or delay the apoptosis of RGCs independent of IOP [9]. Although it may be difficult to identify a single causative factor for the development of glaucoma, a reasonable approach to tackling glaucomatous optic neuropathy remains targeting possible underlying mechanisms of glaucomatous damage, including the deprivation of neurotrophic factors (NTFs), the formation of reactive oxygen species (ROS), oxidative stress, glutamate excitotoxicity (Figure 2), ischemia, glial activation, and genetic determinants [10]. Therefore, understanding pathogenic factors in glaucoma may further pave the way to the development of more practical neuroprotective methods and subsequent clinical translation.

## 2. Neurotrophic Factors

Neurotrophic factors exert a variety of actions by binding to specific receptors and affecting neuron development, survival, and repair via tyrosine kinase signaling [11,12]. Neurotrophic factors are neuroprotective, encouraging axon regeneration and improving neuronal cell function [13]. Because of their promising outcomes in other neurodegenerative disorders of the central nervous system [14,15,16], NTFs are an appealing therapeutic target to explore in glaucoma.

Several NTFs have been reported to be associated with glaucoma, including nerve growth factor (NGF), brain-derived neurotrophic factor (BDNF), ciliary neurotrophic factor (CNTF), fibroblast growth factor 2 (bFGF), glial-derived neurotrophic factor (GDNF), and neurturin [17,18,19,20,21]. These NTFs have been proven in several glaucoma rodent models to be effective in preventing RGC cell death [17,18,19,21]. Clinically, topical treatment with NGF for 3 months was shown to enhance optic nerve functions such as visual field, visual acuity, and contrast sensitivity in a small case series of patients with advanced glaucoma [19]. Most recently, a phase 1b clinical trial evaluating the safety and efficacy profiles of recombinant human NGF eyedrops in glaucoma patients for 8 weeks revealed neither major adverse events nor statistically significant short-term neuroenhancement in terms of structural and functional measures [22]. Such inconsistency in the results may be attributed to the treatment duration as the regeneration of RGCs may need a longer time for observable neuroprotective effects. Nevertheless, based on the good safety profile demonstrated by clinical trials, we could still expect a potential neuroprotective effect if the treatment duration is designed to extend beyond 3 months.

Brain-derived neurotrophic factor enhances RGC survival by activating the extracellular signal-regulated kinases (Erk) Erk1/2 and c-jun, as well as inhibiting caspase 2 [23]. Recently, Cha et al. found that BDNF levels in serum and aqueous humor (AH) were significantly lower in patients with normal tension glaucoma (NTG) and primary open angle glaucoma (POAG) [24]. Clinical research conducted by Oddone et al. showed reduced serum levels of both BDNF and NGF in patients with early to moderate stages of glaucoma [20]. Uzel et al. also observed that BDNF in both serum and AH was lower in POAG patients, and the serum level of BDNF increased significantly three months after trabeculectomy [25]. These results suggest and reinforce the association between BDNF and glaucoma and that BDNF may serve as a potential biomarker for glaucoma detection and disease evaluation. In addition, several prior murine models also demonstrated that BDNF protects and facilitates RGCs’ survival [26,27]. Most recently, Lazaldin et al. found that the intravitreal injection of BDNF can hinder RGC death caused by amyloid-β induced apoptosis in rats [28]. Nevertheless, more efforts are required to reach conclusions about the causal relationship between BDNF and glaucoma as well as whether the supplementation of BDNF is effective as a neuroprotective therapy for glaucoma. 

The CNTF is expressed locally by RGCs. The concentration of CNTF is reduced in the AH and lacrimal fluid in patients with POAG [29]. NT-501, a polymeric device containing a genetically engineered human cell line that secretes CNTF and can be surgically implanted beneath the pars plana, has been trialed in retinal diseases without obvious treatment benefits [30,31]. Clinical trials of NT-501-encapsulated cell therapy are currently being undertaken to explore its therapeutic efficacy in the treatment of glaucoma (Clinical Trials ID NCT02862938 and NCT04577300).

While studies on NTFs show that they have great potential for neuroprotection, the challenge of clinical translation remains in how to accomplish effective and sustainable delivery to the retina. Intravitreal injection is a feasible method of delivering pure recombinant trophic factors to the retina, but in chronic diseases like glaucoma, which may last for several decades, this treatment modality may not be pragmatic if the injection has to be repeated frequently. Therefore, future studies may be directed toward the development of the sustained release of the intraocular implant containing NTFs or stem cell transplant that can modulate the levels of NTFs in the microenvironment. Alternatively, external therapies such as low-level electrical stimulation, in which the electrodes are attached around the eye and on the retina, can be used to induce local production of NTFs [32,33,34]. Previous rodent models have demonstrated the upregulation of CNTF and BDNF after electrical stimulation [35,36]. 

## 3. Ginkgo Biloba

Ginkgo is a traditional medication commonly used in Eastern countries. Modern research has been conducted to explore the neuroprotective effect of Ginkgo biloba extract (GBE) in the treatment of glaucoma, and several theories regarding the underlying mechanisms have been proposed. First, GBE may lead to increased blood flow by altering the blood viscosity and suppressing platelet-activating factors that can induce platelet aggregation, neutrophil degranulation, and ROS generation [37,38,39]. Second, the antioxidative capabilities of GBE may be exerted by its component poly-phenolic flavivonoids via the mechanism of reducing the oxidative stress in the mitochondria and scavenging free radicals [40,41,42,43]. Conflicting clinical results for the effectiveness of GBE have been reported in some clinical trials. Quaranta et al. reported improvements in preexisting visual field defects (average baseline mean deviation of 11.40 dB versus 8.78 dB after GBE treatment, *P* = 0.0001; average baseline-corrected pattern standard deviation of 10.93 dB versus 8.13 dB after treatment, *P* = 0.0001) after one month of 40 mg GBE oral capsule three times daily in patients with NTG without any topical hypotensive therapy [44]. Long-term treatment of 80 mg GBE twice daily also significantly slowed the progression of visual field defects, which improved from the pretreatment regression coefficient of mean deviation at −0.619 dB to −0.379 dB per year without affecting the IOP in NTG patients who concurrently used 1 or 2 hypotensive eyedrops [45]. However, a randomized controlled trial (RCT) revealed no effect of 40 mg GBE tablets three times daily for 4 weeks on visual field performance in NTG patients with insignificant IOP difference during the study period [46]. Although this RCT followed a similar study design to Quaranta’s, the contradictory results may be attributed to some factors such as race and disease severity. In addition, the difference between Lee’s and Guo’s work cannot be compared directly due to different GBE dosages and treatment duration. Recently, Sabaner et al. demonstrated increased peripapillary vessel density on optical coherence tomography (OCT) angiography in healthy subjects after the four-week consumption of GBE 120mg oral capsule [47]. Because previous studies have demonstrated a positive correlation between peripapillary vessel density and visual field performance [48,49], a comprehensive research may be worth conducting to directly evaluate the changes in both vessel density and visual field in patients treated with GBE. Whether GBE is effective in the treatment of glaucoma may still need to be justified with further large clinical trials to identify patient characteristics, disease extent, concurrent treatments, etc. associated with the beneficial effects of GBE in patients with NTG.

## 4. Brimonidine

Aside from its well-known effect of decreasing IOP, brimonidine has shown neuroprotective properties against RGC death in several preclinical studies [50,51,52,53]. It has been shown to increase NTFs and alter N-methyl-D-aspartate (NMDA) receptors involved in glutamate toxicity [54,55]. It is reported to boost BDNF expression in RGCs, resulting in a neuroprotective effect [54]. In addition, brimonidine has been shown to exert neuroprotection by interfering with the amyloid-β pathway and lowering its levels since the disruption of amyloid precursor protein homeostasis and the subsequent accumulation of amyloid-β and its cytotoxicity may contribute to the death of the RGCs [56,57]. In clinical studies, brimonidine monotherapy has been shown to reduce the incidence of visual field progression compared with timolol in treated individuals (9% versus 30%) in the Low Pressure Glaucoma Study Group over 30 months, despite identical IOP-lowering effects [58]. However, the study was restricted by high dropout rates of 55% (54/99) in the brimonidine group and 29% (23/79) in the timolol group, as well as a relatively small sample size involved in the final analysis [59,60]. Topical brimonidine 0.2% administered over 3 months was also observed to increase contrast sensitivity; however, treatment with timolol 0.5% had no benefit, despite identical IOP reduction effects [61]. In addition, Tsai et al. reported no significant change in retinal nerve fiber layer (RNFL) thickness after the administration of brimonidine 0.2 % versus a statistically significant decrease in average RNFL thickness (*P* = 0.004) from baseline in the timolol 0.5 % group in patients with ocular hypertension treated for 1 year, despite similar mean IOP reduction in both groups [62]. Overall, these findings imply that brimonidine has a neuroprotective effect that is not related to IOP and may be used more widely for its IOP-independent treatment effect in glaucoma.

## 5. Calcium Channel Blocker (CCB)

Over the past few decades, calcium dysregulation has been regarded as a pathophysiological component in the degeneration of RGCs [63]. Theoretically, CCBs protect RGCs by preventing cell death caused by calcium influx and increasing local blood flow in ischemic tissues by inducing vasodilation [63,64]. A cell culture study by Yamada et al. demonstrated that CCBs, including iganidipine, nimodipine, and lomerizine, can facilitate RGC viability to sustain hypoxic damage by blocking calcium ion influx into RGCs [65]. Meanwhile, another in vitro study also showed that nilvadipine may be capable of inhibiting glutamate-induced RGC apoptosis by interfering with calcium influx [66]. An RCT revealed that in systemically healthy patients with NTG, nilvadipine in a dosage of 2 mg twice daily may preserve the optic nerve structure as assessed by direct ophthalmoscopy, improve optic nerve head blood flow, and slow visual field progression compared with the placebo group [67]. Another member of CCB, brovincamine, has also been shown to be beneficial in improving visual field results and retarding disease progression in patients with NTG [68,69]. Recently, Duan et al. reported an improvement in ocular hemodynamics as well as visual field defects by nimodipine combined with latanoprost in OAG patients [70]. An increase in superficial macular capillary vessel density was also found in patients with NTG after consuming sixty mg of nimodipine for three months [71]. Despite the aforementioned benefits of CCB revealed in these small-scale studies, one should keep in mind that the influence of vasodilation induced by CCB may not sufficiently explain the neuroprotective effect because vasodilation may inversely direct blood flow away from the ischemic tissues and exacerbate the condition [72]. In addition, reduced systemic blood pressure appears to decrease ONH blood flow and cause further damage to the optic nerve in patients with glaucoma [73]. As a result, future studies may be directed at evaluating the optimal dosage and improving the selectiveness of CCB to exert maximal neuroprotection while minimizing accompanying side effects.

## 6. Memantine

Glutamate is a neurotransmitter activating proapoptotic cascades via NMDA and non-NMDA receptors. Increased glutamate level has been thought to be a possible cause of glaucoma [74]. Memantine is an NMDA receptor antagonist that inhibits excessive glutamate activity, has been studied in experimental models of glaucoma, and has demonstrated a protective effect on RGC survival and reductions in functional loss [74,75,76,77]. Despite promising results from animal models, a large phase 3 RCT showed that daily treatment of memantine over 4 years has little impact on delaying visual field progression in patients with bilateral open-angle glaucoma [78]. Several variables influencing therapeutic effectiveness may be taken into account to explain the discrepancy between animal studies and clinical trials, including baseline glaucoma severity, trial duration, memantine dosage, and administration route. In addition, because glaucoma is a multifactorial disease, such complexity may be impossible to simulate by a single animal model. Additional factors that could affect the treatment outcome should also be evaluated, including the presence of disc hemorrhage, central corneal thickness, etc. Further research may be directed toward enrolling patients with early-stage glaucoma, a less heterogeneous population in terms of different glaucoma progression risk factors, longer study duration, and other doses and methods for medication delivery. Although the clinical study failed to prove the association between memantine and visual field outcome, studies investigating the changes in RNFL using OCT and OCT angiography are also warranted as they represent more objective structural parameters in the assessment and detection of early glaucoma [79,80].

## 7. Citicoline

Citicoline is a natural compound that participates in the chemical reaction involving the neurotransmitter acetylcholine and other neuronal membrane components. It is crucial for preserving the levels of sphingomyelin and cardiolipin in neurons [81]. Citicoline exerts its neuroprotective effect by reducing glutamate excitotoxicity, lowering oxidative stress in RGC damage, and improving axonal transport deficit [82,83]. While the axons of RGC in the retrobulbar space are rich in myelin, citicoline may be a potential treatment option to modulate RGC viability via phospholipid metabolism in the myelin membrane [84].

In an animal model of glaucoma using adult rats with optic nerve crush, a decrease in RGC density was attenuated following intraperitoneal citicoline administration, indicating a protective effect against neuronal degeneration [85]. Improvements in pattern electroretinogram (PERG) and visual evoked potentials (VEP) were demonstrated in an RCT enrolling participants with POAG following the intramuscular injection of citicoline 1000 mg/day for 60 days [86]. Similar findings were also observed in glaucoma patients treated with topical citicoline eyedrops [87,88]. A study consisting of 47 POAG patients with beta-blocker monotherapy revealed that patients treated with topical citicoline (OMK1^®^, Omikron Italia, 3 drops/day) over 4 months had improved PERG and VEP, whereas no significant changes in PERG and VEP were observed in the control group treated with beta-blocker monotherapy [87]. In addition, 4 cycles of treatment, in which each cycle contains oral citicoline solution 1 vial (500 mg of citicoline) for 4 months and stopped for 2 months, were effective in halting visual field progression in 41 patients with POAG despite well-controlled IOP with hypotensive medications [89]. Recently, an RCT revealed that the addition of citicoline eyedrops 3 times daily for 3 years in patients with progressing OAG might be effective in terms of reducing the progression of mean deviation on 10-2 visual field and RNFL thickness [90]. Based on the aforementioned findings, it appears that citicoline offers great potential as a future therapeutic strategy for glaucoma and other neurological illnesses. Several trials are currently underway to explore the treatment effect of citicoline and its related products. (Clinical Trials ID NCT05315206, NCT04499157, NCT04784234).

## 8. Antioxidant Q10

Coenzyme Q10 (CoQ10) is an important endogenous antioxidant and electron transport chain component [91]. The intraocular administration of CoQ10 therapies has been demonstrated to help delay RGC apoptosis and reduce glutamate concentration in a rat model [92]. A diet containing CoQ10 has also been shown to be neuroprotective against NMDA-induced retinal damage both in vitro and in mice in vivo [93]. In another mouse model of glaucoma, a diet supplemented with CoQ10 could reduce glutamate excitotoxicity and oxidative stress-mediated RGC degeneration and improve RGC survival by 29% [94]. A similar neuroprotective effect was also observed in a rat model given topical CoQ10 and vitamin E treatment for 4 weeks after mechanic optic nerve injury [95]. A small study conducted by Parisi et al. revealed that in patients with POAG, the administration of CoQ10 plus vitamin E had a beneficial effect on the inner retinal function and visual cortical responses by using PERG and VEP, respectively [96]. In an RCT containing 64 eyes with pseudoexfoliative glaucoma, a lower level of superoxide dismutase in the AH was found in patients treated with topical CoQun solution containing CoQ10 and vitamin E [97]. Currently, a large multicenter RCT evaluating the neuroprotective effects of CoQun eyedrops in addition to prostaglandin monotherapy on glaucoma progression in patients with POAG is still undergoing (Clinical Trials ID NCT03611530) [98]. As more studies are being conducted, we can expect the treatment potential of CoQ10 in conjunction with IOP-lowering medications in the management of glaucoma.

## 9. Nicotinamide (Vitamin B3)

Nicotinamide, also known as vitamin B3, is a precursor to nicotinamide adenine dinucleotide (NAD), which is a co-enzyme found in living cells and an essential molecule for the proper function of the metabolic system. As reduced NAD levels and mitochondrial dysfunctions are considered to be the hallmarks of the aging process [99], the question remains whether the repletion of nicotinamide could be beneficial in the treatment of neurodegenerative diseases including glaucoma. In 2017, Williams et al. first demonstrated that oral supplementation with high-dose nicotinamide could alleviate the decreased NAD levels in the retina caused by aging in DBA/2 J mice, a mouse model of age-dependent inherited glaucoma [100]. In line with the findings of NAD depletion in the murine model, Nzoughet et al. revealed a reduced plasma level of NAD in patients with POAG compared with the control group [100,101]. This finding further suggests the potential role of nicotinamide supplementation in treating human glaucoma. Therefore, to evaluate the effect of supplementation of nicotinamide in human glaucoma, Hui et al. examined 57 participants with early to moderate glaucoma with well-controlled IOP who received either oral placebo or nicotinamide at the dosage of 1.5 g/day for 6 weeks and 3 g/day for another 6 weeks, and all participants crossed over without washout after 12 weeks. Overall, they found that patients consuming nicotinamide had improved RGC function as evaluated by ERG, independent of IOP [102]. A recent phase 2 RCT also showed a greater number of improving visual field test locations and improved rates of change of pattern standard deviation in patients with treated, moderate open-angle glaucoma taking a combination of oral nicotinamide and pyruvate [103]. These findings emphasize the importance of nicotinamide supplementation in glaucoma treatment, and the ongoing clinical trials examining the long-term effects of high-dose nicotinamide in glaucoma will be crucial in establishing the therapeutic role of NAM supplementation to slow the loss of visual field in people with glaucoma. (Clinical Trials ID NCT05275738, NCT05405868).

## 10. Statins

Statins are medications originally used to treat hypercholesterolemia. Their major mechanism of action is to block HMG-CoA reductase and hence reduce cholesterol production. A recently published meta-analysis of observational studies reported that glaucoma was linked to high total cholesterol and low high-density lipoprotein levels [104], which strengthens the importance of blood lipid levels in the treatment of glaucoma. A rat model of chronic ocular hypertension was used to evaluate the neuroprotective effect of statins and showed the improved survival of RGCs, reduced apoptosis, and the suppression of glial activation in the retina in the statins group [105]. One study using confocal scanning laser ophthalmoscopy to measure the rate of progression of optic nerve parameters in glaucoma suspects taking statin showed a slowed progression of loss of rim volume, cross-sectional area of RNFL, and mean global RNFL thickness when compared with the control group [106]. Despite promising preclinical data, Kang et al. found no significant association between statin use and rates of change in mean deviation and RNFL thickness in those with glaucoma or glaucoma suspects [107]. The most recent meta-analysis revealed a slightly lower risk of OAG onset after using statins, whereas the association between statin use and OAG progression is still uncertain [108]. With the increasing application of statins in the treatment of cardiovascular and cerebrovascular diseases [109,110], more preclinical and clinical studies are warranted to elucidate the underlying neuroprotective mechanisms of statins against glaucomatous optic neuropathy.

## 11. Stem Cell Therapy

Stem cell therapy is gaining in popularity for its potential to treat neurodegenerative diseases such as glaucoma. In glaucoma, the ultimate goal is to restore vision by the neuroregeneration of injured or dead RGCs and their axons. Stem cell treatment may be therapeutic for glaucoma through two different mechanisms: (1) regenerating RGCs and producing new cells of different kinds. (2) providing a favorable neurotrophic environment to the damaged RGCs [111,112]. In addition, the RGCs are the ideal target for stem cell therapy because they have the benefit of being confined to the intraocular spaces and may be less likely to be affected by immune rejection [113]. 

Mesenchymal stem cells (MSCs) are multipotent and can differentiate into neurons and glial cells, support neuronal growth and synaptic connection, induce angiogenesis, modulate inflammatory responses, and reduce demyelination and apoptosis, which all contribute to their neuroprotective and regenerative effects [114]. The transplanted MSCs can secret various NTFs to promote cell survival, including CNTF (a potent RGC survival factor), bFGF (a simulator of axonal growth), GDNF, and BDNF [112,115]. Various IOP-dependent animal models of glaucoma have demonstrated effectiveness in terms of promoting RGC survival and reducing RGC loss via the intravitreal injection of MSCs [116,117,118,119,120] and protecting trabecular meshwork tissue via the intracameral injection of MSCs [121]. Recently, a clinical trial by Vivela et al. reported no significant improvement in visual performance or ERG in a patient with advanced glaucoma after the intravitreal injection of autologous bone marrow-derived MSCs. In addition, the development of retinal detachment with proliferative vitreoretinopathy was noted in another participant [122]. Therefore, despite successful outcomes shown in animal models, there are still obstacles that can hinder the clinical translation of stem cell therapy into human application. Particularly, the complexity of human disease states may not be exactly represented by a controlled experimental environment in animal models. Nevertheless, larger clinical trials enrolling more participants with different disease severity; using different administration routes, i.e., intracamerally or intravitreally; and following for a longer period of duration are still warranted to fully elucidate the clinical effectiveness of this treatment modality.

Some safety issues need to be addressed before stem cell therapy could be successfully used in clinical practice. First, the balance between graft survival and tumorigenesis must be carefully assessed because the longer the stem cell lasts, the more likely the tumor might develop. Thus, meticulous laboratory and clinical research will be required to guarantee that the potential benefit of neuroprotection far outperforms the possible danger of tumor induction. Second, the implanted cells not only release ideal and desired trophic factors for supporting RGCs, but they may also secret other agents that could be potentially detrimental to the microenvironment of RGC [123]. Third, the differences in efficacy among various animal models for glaucomatous optic neuropathy may need to be justified before clinical translation. Therefore, more efforts are required before stem cell therapy can become practical in clinical settings.

## 12. Gene Therapy

Gene therapy has made remarkable progress in the past few decades. It offers the potential to help patients with damaged RGCs regain their lost vision. Clinical trials conducted on patients with congenital retinal diseases such as Leber hereditary optic neuropathy have shown promising results with direct gene medical application in the treatment of optic neuropathy [124,125]. However, genetic treatment in the context of glaucoma remains challenging due to its multifactorial and polygenic properties. 

Some animal studies have demonstrated the efficacy of gene therapy in the treatment of glaucoma. For example, experimental studies conducted by Jain et al. showed that clustered regularly interspaced short palindromic repeats (CRISPR)-mediated genome editing of myocilin (MYOC)-dominant gain-of-function mutations effectively lowered IOP and hindered glaucomatous damage by inducing the loss of function of mutant MYOC in a mouse model of MYOC-associated POAG [126]. Another promising result was shown in a recent study using multiple rodent glaucoma models, in which the reactivation of CaMKII activity via the intravitreal injection of adeno-associated virus (AAV) vectors in diseased mice protects RGCs and preserves visual function and visually guided behavior [114]. Gene therapy also exerts its neuroprotective effect through encoding NTFs such as BDNF and CNTF [127,128]. In a rat model of optic nerve injury, Osborne et al. demonstrated enhancements of RGC survival and no significant adverse effects on the retinal structure or electrophysiological performance following the intravitreal injection of AAV2 TrkB-2A-mBDNF [127]. Various gene targets have also been studied in the experimental models of glaucoma, including BCLXL, NMNAT2, Myc-associated protein X, and XIAP [129,130,131,132]. 

With the advancement of whole-genome sequencing and genome editing technology, further genes related to the pathogenesis of glaucoma will be able to be discovered and tested as potential therapeutic targets. While there are still many obstacles to overcome before glaucoma gene therapy becomes clinically available, the progress in understanding the genetic etiology of glaucoma and breakthroughs in RGC neuroprotection in various animal models still hold the potential for leading to a new frontier of gene therapy in glaucoma. 

## 13. Conclusions

Neuroprotection has the potential to play a critical role in glaucoma treatment. Improvement in RGCs survival and a decrease in cell death can not only slow disease progression but even restore visual function through tissue regeneration. Although several treatment modalities exhibit neuroprotective effects in experimental or clinical studies regarding glaucoma, only a few of them have resulted in approved therapy clinically, and the road to glaucoma neuroprotection remains long. Appendix A summarizes the essence of the clinical study of each treatment modality and its implications. Overall, in clinical scenarios such as patients who are intolerant to hypotensive medications or unwilling to receive laser or surgical intervention or who have progressive glaucomatous defect despite well-controlled IOP, supplements such as GBE, citicoline, antioxidant Q10, and nicotinamide may be considered if they are available and not harmful to the patient’s health. Brimonidine may also be used at the physician’s discretion. The clinical usage and effectiveness of CCB, memantine, and statin remain to be justified. Neurotrophic factors, stem cell therapy, and gene therapy warrant further investigation before they can be administered to patients. Advancement in the evolution of neuroprotective therapy will be aided by substantial investment in genetic and biomolecular research.

## Figures and Tables

**Figure 1 jpm-12-01884-f001:**
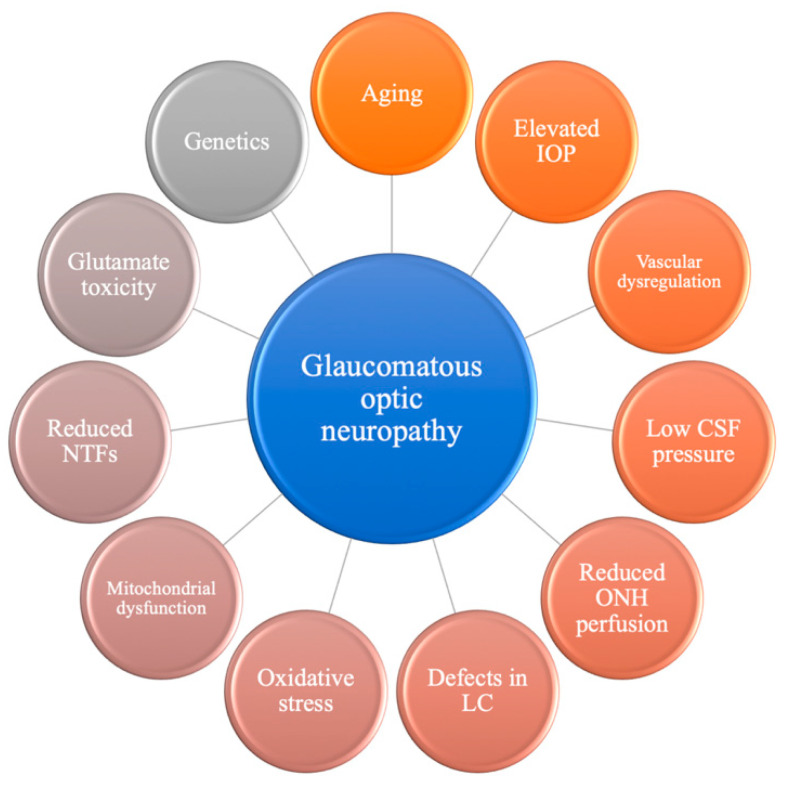
Overview of multifactorial mechanisms contributing to the development of glaucomatous optic neuropathy. CSF, cerebrospinal fluid; IOP, intraocular pressure; LC, lamina cribrosa; NTFs, neurotrophins; ONH, optic nerve head.

**Figure 2 jpm-12-01884-f002:**
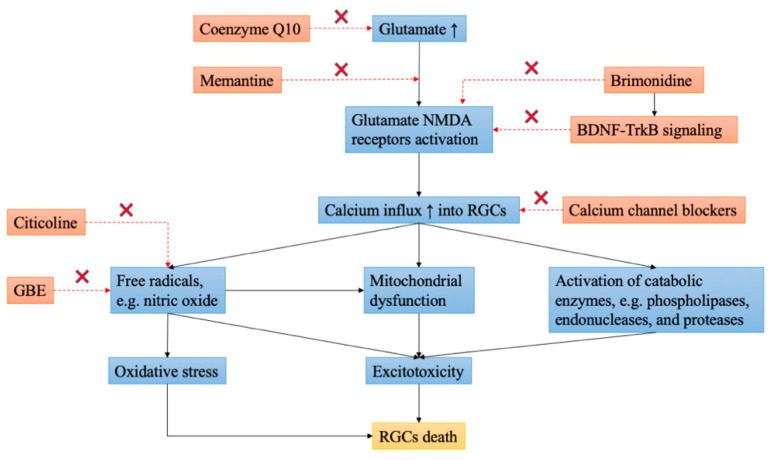
Schematic diagram illustrating the mechanism of glutamate excitotoxicity and summarizing potential therapeutic mechanisms of several agents in neuroprotection related to glutamate excitotoxicity. The symbol “×” indicates inhibition. BDNF, brain-derived neurotrophic factor; GBE, ginkgo biloba extract; NMDA, N-methyl-D-aspartate; RGCs, retinal ganglion cells; TrkB, tropomyosin receptor kinase B.

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
