# Peer review of "Neuroprotection in Glaucoma: Basic Aspects and Clinical Relevance"

_jpm, 2022, doi:10.3390/jpm12111884_

Round 1
Reviewer 1 Report
In this manuscript, the author summarized the potential neuroprotective therapied for glaucoma treatment. I only have 3 comments and I hope the authors can address them.
1. The author may also consider to add 1 table to summarize the potential neuroprotective therapies. The table could include the mechanism behind the treatment, clinical trials et al. Especially the clinical trial information can be very useful and straightforward.
2. Line 88~94, the author first indicated that the clinical treatment of glaucoma with NGF eyedrops have shown to enhance the nerve function. Then the author further indicated “in phase I clinical trials, there is no statistically significant short-term neuroenhancement in terms of structural and functional measures”. Please give some comments on such inconsistency. Additional comments could be helpful for readers to have some take-away notes.
3. For the gene therapy section, please add more comments about the potential gene and cell targets for the glaucoma treatment. Actually CRISPER, AAV even LNP are mainly worked as the carrier to effective delivery the gene to the target cells.
Author Response
Dear editors and reviewers,
We appreciate the time and effort you have dedicated to providing your valuable feedback on our manuscript. On behalf of all the authors of this manuscript “Neuroprotection in glaucoma: basic aspects and clinical relevance " (Manuscript ID jpm-1950378), I would like to submit our responses to the reviewers’ comments and suggestions as follows:
Comment 1: The author may also consider to add 1 table to summarize the potential neuroprotective therapies. The table could include the mechanism behind the treatment, clinical trials et al. Especially the clinical trial information can be very useful and straightforward.
Response 1: Thank you very much for your valuable comments and suggestions on our manuscript. the information on each treatment modality and the clinical trial is listed in Table 1.
Comment 2: Line 88~94, the author first indicated that the clinical treatment of glaucoma with NGF eyedrops have shown to enhance the nerve function. Then the author further indicated “in phase I clinical trials, there is no statistically significant short-term neuroenhancement in terms of structural and functional measures”. Please give some comments on such inconsistency. Additional comments could be helpful for readers to have some take-away notes.
Response 2: Thank you for your suggestions. The additional comments were added on page 3, line 93-97, “Such inconsistency in the results may be attributed to the treatment duration as regeneration of RGCs may need a longer time for observable neuroprotective effects. Nevertheless, based on the good safety profile demonstrated by clinical trials, we could still expect a potential neuroprotective effect if the treatment duration is designed to extend beyond 3 months.”
Comment 3: For the gene therapy section, please add more comments about the potential gene and cell targets for the glaucoma treatment. Actually CRISPER, AAV even LNP are mainly worked as the carrier to effective delivery the gene to the target cells.
Response 3: We appreciate the reviewer’s comment. Additional comments and information were added to the revised manuscript on page 9, line 396-401, “Various gene targets have also been studied in the experimental models of glaucoma, including BCLXL, NMNAT2, Myc-associated protein X, and XIAP [129-132].
With the advancement of whole-genome sequencing and genome editing technology, further genes related to the pathogenesis of glaucoma would be able to be dis-covered and tested as potential therapeutic targets.”.
Corresponding Author
Catherine Jui-Ling Liu, MD.
Department of Ophthalmology, Taipei Veterans General Hospital,
No. 201, Sec.2, Shih-Pai Road, Taipei, Taiwan
E-mail: jlliu@vghtpe.gov.tw; Tel: 886-2-28757325
Reviewer 2 Report
The authors present a review of the literature regarding various clinically-relevant treatment strategies that have been demonstrated to provide neuroprotection in glaucoma My comments and suggestions for improvement are as follows:
1) There is a major area of research that is missing from this review, which is that of vitamin supplementation. In particular, there is growing evidence that nicotinamide (vitamin B3) has potential to improve inner retinal function in glaucoma (https://pubmed.ncbi.nlm.nih.gov/32721104/). Similarly, other researchers have identified in clinical trials that a combination of nicotinamide and pyruvate plays a role in neuroprotection (https://pubmed.ncbi.nlm.nih.gov/34792559/). This is a major area of emerging research that needs to be covered.
2) Although each treatment strategy is covered in good detail, the conclusion at the end does not sufficiently summarise the paper. There needs to be more emphasis on what value this paper adds to the literature too. It would be good to see more in-depth discussion at the end summarizing the various options and discussing which strategies are most promising relative to others that were covered in the paper. Perhaps a summary comparison table including information such as efficacy, mechanism of action and limitations would be helpful for readers.
Author Response
Dear editors and reviewers,
We appreciate the time and effort you have dedicated to providing your valuable feedback on our manuscript. On behalf of all the authors of this manuscript “Neuroprotection in glaucoma: basic aspects and clinical relevance " (Manuscript ID jpm-1950378), I would like to submit our responses to the reviewers’ comments and suggestions as follows:
Comment 1: There is a major area of research that is missing from this review, which is that of vitamin supplementation. In particular, there is growing evidence that nicotinamide (vitamin B3) has potential to improve inner retinal function in glaucoma (https://pubmed.ncbi.nlm.nih.gov/32721104/). Similarly, other researchers have identified in clinical trials that a combination of nicotinamide and pyruvate plays a role in neuroprotection (https://pubmed.ncbi.nlm.nih.gov/34792559/). This is a major area of emerging research that needs to be covered.
Response 1: Thank you for your suggestions. We added a paragraph to summarize and discuss the important role of vitamin B3 in the neuroprotection of glaucoma.
Comment 2: Although each treatment strategy is covered in good detail, the conclusion at the end does not sufficiently summarise the paper. There needs to be more emphasis on what value this paper adds to the literature too. It would be good to see more in-depth discussion at the end summarizing the various options and discussing which strategies are most promising relative to others that were covered in the paper. Perhaps a summary comparison table including information such as efficacy, mechanism of action and limitations would be helpful for readers.
Response 2: We appreciate the reviewer’s insightful comment and suggestion. The conclusion was amended. In addition, the key points and clinical implications of each study are summarized in Table 1.
Corresponding Author
Catherine Jui-Ling Liu, MD.
Department of Ophthalmology, Taipei Veterans General Hospital,
No. 201, Sec.2, Shih-Pai Road, Taipei, Taiwan
E-mail: jlliu@vghtpe.gov.tw; Tel: 886-2-28757325
Reviewer 3 Report
This article lists different neuroprotective to promote RGC survival in glaucoma patients. The work gives a idea of the mechanisms that leads to the death of the RGC, as well as the different points where it can act. This general vision, with the specific approaches, makes it worthy of being published. However, there are a number of points to consider:
Figure 2 represents the abstract of the work, so it should be cited in the text as such and not simply to point to glutamate excitotoxicity. On the other hand, the scheme of the figure suggests the structure of the article, so the article would improve by pointing out the possible places where neuroprotection has been tried, and has some success, and giving the indicated groups as an example, rather than a mere list of factors and substances proven to achieve said neuroprotection.
Between lines 88 and 94 they indicate two contradictory results in relation to topical treatment with NGF, they should be discussed.
BDNF is a recognized trophic factor, however it is presented more as a marker as it decreases when glaucoma appears and increases when a trabeculectomy is performed. It seems more a consequence than a cause of neuroprotection. In addition, a further explanation of where and how electrical stimulation should be done to induce local production of NTFs and what results have been obtained would be appreciated.
They present the disparity of results after the application of Ginkgo Biloba in tablets or in eye drops (lines 139-143 with respect to 143-145), a possible justification for this discrepancy would be appreciated.
On the other hand, regarding the work of Sabaner, what functional consequences would the increase in peripapillary vascular density have after the consumption of GBE?
A comparison is made of the effects that the different neuroprotectors have. However, since they seem to act in different ways, the comparison would be easier if, in addition to the text, a table was presented comparing the effects (pros and cons of each one) to choose the most appropriate in each situation or the combination of the more suitable.
In the case of memantine, the cause of the differences between clinical trials and animal models should be further discussed, and even between the same animal models, it is not enough to state these differences.
RGCs are said to be an ideal target for stem cells because they are confined to the intraocular space beyond the reach of the immune system (line 296-298). However, they are targeted by microglia when the process of apoptosis begins. Therefore, the sentence should be made a little more precise. On the other hand, differences are also shown between the results obtained in animal models and humans. They should discuss the reasons why this therapy works in animal models and not in humans. What are the parameters to take into account that change between both models?
Author Response
Dear editors and reviewers,
We appreciate the time and effort you have dedicated to providing your valuable feedback on our manuscript. On behalf of all the authors of this manuscript “Neuroprotection in glaucoma: basic aspects and clinical relevance " (Manuscript ID jpm-1950378), I would like to submit our responses to the reviewers’ comments and suggestions as follows:
Comment 1: This article lists different neuroprotective to promote RGC survival in glaucoma patients. The work gives an idea of the mechanisms that leads to the death of the RGC, as well as the different points where it can act. This general vision, with the specific approaches, makes it worthy of being published. However, there are a number of points to consider:
Figure 2 represents the abstract of the work, so it should be cited in the text as such and not simply to point to glutamate excitotoxicity. On the other hand, the scheme of the figure suggests the structure of the article, so the article would improve by pointing out the possible places where neuroprotection has been tried, and has some success, and giving the indicated groups as an example, rather than a mere list of factors and substances proven to achieve said neuroprotection.
Response 1: Thank you for your comment. Although the initiative of drafting figure 2 was to additionally summarize the possible role that glutamate plays in most of the treatment modalities instead of suggesting the glutamate excitotoxicity as the backbone of the whole manuscript, we still really appreciate the reviewer’s suggestions for the manuscript. The manuscript has been thoroughly revised.
Comment 2: Between lines 88 and 94 they indicate two contradictory results in relation to topical treatment with NGF, they should be discussed.
Response 2: We appreciate the reviewer’s insightful comment. The additional comments were added on page 3, line 93-97, “Such inconsistency in the results may be attributed to the treatment duration as regeneration of RGCs may need a longer time for observable neuroprotective effects. Nevertheless, based on the good safety profile demonstrated by clinical trials, we could still expect a potential neuroprotective effect if the treatment duration is designed to extend beyond 3 months.”
Comment 3: BDNF is a recognized trophic factor, however it is presented more as a marker as it decreases when glaucoma appears and increases when a trabeculectomy is performed. It seems more a consequence than a cause of neuroprotection.
Response 3: The reviewer is right to point out that there is a lack of evidence to suggest the role of BDNF as a neuroprotective agent in terms of treatment in the current manuscript. We added the information regarding past and the most updated in vivo study results to emphasize the potential treatment effect of BDNF on page 4, line 108-110, “In addition, several prior murine models also demonstrated that BDNF protects and facilitates RGCs' survival [26,27]. Most recently, Lazaldin et al. found intravitreal injection of BDNF can hinder RGC death caused by amyloid-β induced apoptosis in rats [28].”
Comment 4: In addition, a further explanation of where and how electrical stimulation should be done to induce local production of NTFs and what results have been obtained would be appreciated.
Response 4: We appreciate the reviewer’s comment. The additional information was added on page 4, line 128-132, “Alternatively, external therapies such as low-level electrical stimulation, in which the electrodes are attached around the eye and on the retina, can be used to induce local production of NTFs [32-34]. Previous rodent models have demonstrated upregulation of CNTF and BDNF after electrical stimulation [35,36].”
Comment 5: They present the disparity of results after the application of Ginkgo Biloba in tablets or in eye drops (lines 139-143 with respect to 143-145), a possible justification for this discrepancy would be appreciated.
Response 5: Thank you for your comment. The additional comments were added on page 4-5, line 153-157, “Although this RCT followed a similar study design to Quaranta’s, the contradictory results may be attributed to some factors such as race and disease severity. In addition, the difference between Lee’s and Guo’s work cannot be compared directly due to different GBE dosages and treatment duration.”
Comment 6: On the other hand, regarding the work of Sabaner, what functional consequences would the increase in peripapillary vascular density have after the consumption of GBE?
Response 6: Thank you for your comment. The additional comments were added on page 5, line 159-162, “Because previous studies have demonstrated a positive correlation between peripapillary vessel density and visual field performance [48,49], comprehensive research may be worth conducting to directly evaluate the change of both vessel density and visual field in patients treated with GBE.”
Comment 7: A comparison is made of the effects that the different neuroprotectors have. However, since they seem to act in different ways, the comparison would be easier if, in addition to the text, a table was presented comparing the effects (pros and cons of each one) to choose the most appropriate in each situation or the combination of the more suitable.
Response 7: Thank you for your suggestion. The key points and clinical implications of each study are summarized in Table 1.
Comment 8: In the case of memantine, the cause of the differences between clinical trials and animal models should be further discussed, and even between the same animal models, it is not enough to state these differences.
Response 8: Thank you for bringing up these issues. We amended the paragraph and added additional comments on the cause of the differences between clinical trials and animal models.
Comment 9: RGCs are said to be an ideal target for stem cells because they are confined to the intraocular space beyond the reach of the immune system (line 296-298). However, they are targeted by microglia when the process of apoptosis begins. Therefore, the sentence should be made a little more precise. On the other hand, differences are also shown between the results obtained in animal models and humans. They should discuss the reasons why this therapy works in animal models and not in humans. What are the parameters to take into account that change between both models?
Response 9: We appreciate the reviewer’s insightful comment. The first issue was amended as, “the RGCs are the ideal target for stem cell therapy because they have the benefit of being confined to the intraocular spaces and may be less likely to be affected by immune rejection[113]” on page 8, line 342-344. Additional comments for the second issue were added on page 8-9, line 357-364, “Therefore, despite successful outcomes shown in animal models, there are still obstacles that can hinder the clinical translation of stem cell therapy into the human application. Particularly, the complexity of human disease states may not be exactly represented by a controlled experimental environment in animal models. Nevertheless, larger clinical trials enrolling more participants with different disease severity, using different administration routes i.e. intracamerally or intravitreally, and following for a longer period of duration are still warranted to fully elucidate the clinical effectiveness of this treatment modality.”
Corresponding Author
Catherine Jui-Ling Liu, MD.
Department of Ophthalmology, Taipei Veterans General Hospital,
No. 201, Sec.2, Shih-Pai Road, Taipei, Taiwan
E-mail: jlliu@vghtpe.gov.tw; Tel: 886-2-28757325
Round 2
Reviewer 2 Report
I thank the authors for their response to my initial comments and suggestions. I have a few comments that remain as follows:
1) Although the authors have provided a table that summarises the findings for each treatment modality, the information provided is for each treatment in isolation and does not provide the reader with any comparisons across treatments. Strengths and drawbacks with respect to the other available options would help provide some context in terms of which options are most promising.
2) It is good to see that the authors have included a summary paragraph outlining which treatment options are preferred. However, there is no justification or rationale provided in terms of why the authors have identified certain treatment options as having the greatest efficacy. Currently it is not entirely clear how the authors have arrived at their conclusion.
3) Emphasis needs to be made on what additional value this paper adds to the literature and what differentiates this from other published reviews on neuroprotection in glaucoma that are already available.